# Dietary Patterns in Relation to Prospective Sleep Duration and Timing among Mexico City Adolescents

**DOI:** 10.3390/nu12082305

**Published:** 2020-07-31

**Authors:** Erica C. Jansen, Ana Baylin, Alejandra Cantoral, Martha María Téllez Rojo, Helen J. Burgess, Louise M. O’Brien, Libni Torres Olascoaga, Karen E. Peterson

**Affiliations:** 1Department of Nutritional Sciences, School of Public Health, University of Michigan, Ann Arbor, MI 48109, USA; abaylin@umich.edu (A.B.); karenep@umich.edu (K.E.P.); 2Division of Sleep Medicine, University of Michigan, Ann Arbor, MI 48109, USA; louiseo@med.umich.edu; 3Department of Epidemiology, School of Public Health, University of Michigan, Ann Arbor, MI 48109, USA; 4CONACYT, National Institute of Public Health, Cuernavaca 62000, Mexico; alejandra.cantoral@insp.mx; 5Center for Research on Nutrition and Health, National Institute of Public Health, Cuernavaca 62000, Mexico; mmtellez@insp.mx (M.M.T.R.); libniavib@gmail.com (L.T.O.); 6Department of Psychiatry, University of Michigan, Ann Arbor, MI 48109, USA; bhelen@med.umich.edu; 7Department of Obstetrics and Gynecology, University of Michigan, Ann Arbor, MI 48109, USA

**Keywords:** fruits, vegetables, lean proteins, sleep health, circadian

## Abstract

Adult studies show that healthy diet patterns relate to better sleep. However, evidence during adolescence, when sleep may change dramatically, is lacking. Within a cohort of 458 Mexican adolescents, we examined whether consumption of three dietary patterns was associated with sleep duration and timing measured 2 years later, as well as changes in sleep timing and duration. Dietary patterns (identified a posteriori in a prior analysis) were assessed with a baseline food frequency questionnaire, and sleep was measured with wrist actigraphy at baseline and follow-up. Linear regression analyses adjusting for sex, age, screen time, and smoking were conducted. Adolescents with higher consumption of a Plant-Based and Lean Proteins pattern had earlier sleep timing (−0.45 h with 95% Confidence Interval (CI) −0.81, −0.08 in the highest compared to lowest quartiles), less of a phase delay in sleep timing over follow-up (−0.39 h with 95% CI −0.80, 0.02), and shorter weekend sleep duration (0.5 h with 95% CI −0.88, −0.1). Higher consumption of an Eggs, Milk and Refined Grain pattern was associated with earlier sleep timing (−0.40 h with 95% CI −0.77, −0.04), while consumption of a Meat and Starchy pattern was related to higher social jetlag (weekend–weekday sleep timing difference). Healthier diet patterns may promote better sleep in adolescents.

## 1. Introduction

Healthy sleep, including adequate sleep duration and earlier sleep timing, is essential for promoting cardiometabolic health among adolescents (individuals aged approximately 10 to 19 years) [1]. Recent evidence shows that both short sleep duration and delayed sleep timing are related to higher adiposity [2,3], insulin resistance [4,5], and blood pressure [6,7]. However, both short sleep duration and late bedtimes are prevalent among adolescents, with over half of adolescents worldwide not achieving the recommended sleep duration [8].

While there are multiple predictors of sleep health among adolescents (e.g., early school start times, technology and social media use, and poor mental health [9]), diet could play an important role. It is well known that sleep and diet are connected, and most likely in a bidirectional manner. Laboratory evidence from adult studies shows that, after a night of sleep deprivation, appetite and consumption patterns are altered [10,11]. Epidemiological evidence from pediatric populations also point to connections between short sleep and higher consumption of lower-quality foods that are heavily processed and high in added sugar [12,13,14]. In contrast, individual foods and nutrients may help to promote optimal sleep health [15]. We have shown that docosahexaenoic acid (DHA), an omega 3 fatty acid obtained primarily through fatty fish, was associated with earlier sleep timing and longer sleep duration on weekends among adolescents [16]. In the adult literature, recent attention has focused on examining dietary patterns as a whole in relation to prospectively assessed sleep. Dietary patterns can refer to those defined a priori through a validated score or defined a posteriori through data-driven methods. Higher adherence to a Mediterranean diet pattern, a validated score, associated with fewer insomnia symptoms in a diverse US population [17] and healthier dietary patterns (identified through data-driven methods) associated with better sleep quality over time in peri-menopausal Mexican women [18].

Less is known about the role of diet patterns in sleep among adolescents, especially from studies with prospective data over the pubertal time frame. The adolescent period is a vulnerable period characterized by hormonal and physiological changes, including changes in sleep timing and duration. In particular, alterations in the timing of the circadian clock (observed in hormones such as melatonin) and sleep pressure (the drive to fall asleep) result in delayed bed and wake times relative to childhood [19]. However, there are variations within adolescents regarding the extent of these changes, with some adolescents experiencing more dramatic phase changes in sleep timing and duration. Whether healthier dietary patterns, particularly dietary patterns that promote melatonin secretion and/or are antioxidative [20], could promote less dramatic changes in sleep timing and duration remains unknown. Thus, the aims of the present study were two-fold: 1) to evaluate whether healthier dietary patterns measured in adolescence were predictive of longer sleep duration and earlier sleep timing assessed two years later, and 2) to examine whether healthier dietary patterns measured in adolescence were associated with lower changes in sleep duration or timing over the two-year follow-up.

## 2. Materials and Methods

The study sample included adolescent participants from two of three sequentially enrolled cohorts of the Early Life Exposure in Mexico to ENvironmental Toxicants (ELEMENT) study [21]. Between 1997 and 2004, 1012 mother/child dyads were recruited from prenatal clinics of the Mexican Social Security Institute in Mexico City, which serves low- to middle-income populations formally employed in the private sector. At baseline, mothers reported sociodemographic and health information. In 2015, a subset of 554 participants from the original birth cohorts 2 and 3, who were in the midst of the pubertal transition (ages 9 to 17 years), were selected to participate in a follow-up visit (called time 1 or T1 in the present study). Starting in 2017, 519 (94%) of the same adolescents participated in an additional follow-up visit that was identical to the 2015 visit (time 2 or T2 in the present study). Overall, there were 458 adolescents who had dietary information at T1 as well as valid wrist actigraphy data at both T1 and T2. Compared to the full sample of participants, the analytic study sample was less likely to have tried smoking (24% versus 38%; *p* = 0.046), but otherwise did not vary with regards to important sociodemographic and lifestyle characteristics. The institutional review boards at the Mexico National Institute of Public Health and the University of Michigan approved the research protocols (CI 599 and HUM00034344). Informed consent was obtained from parents for all participants in addition to participant assent.

### 2.1. Diet

During the adolescent follow-up visit, a trained interviewer administered a 104-item semi-quantitative food frequency questionnaire (FFQ) to the adolescents to obtain information on typical consumption habits. This questionnaire has been validated in a Mexican population [22]. The FFQ asked adolescents to recall how often they had consumed one serving of standard portion sizes of each food item over the previous week. Photographs of food items were provided as a visual aid. Response options ranged from never to ≥6 times per day. We converted the raw response values (1–9) into servings/day and adjusted for total energy intake using the residual method [23]. Using the energy-adjusted frequency data, three dietary patterns were identified in a previous analysis using principal components analysis [24]. The first pattern, called the Plant-Based and Lean Proteins pattern, was characterized by high intake frequency of vegetables, fruit, soup, fish, water and unsweetened drinks, and high-fat dairy. The second pattern, called Meat and Starchy, was marked by high intake frequency of “Western” processed foods including chips, refined grains, sugar-sweetened beverages, processed meat, and high-fat dairy, as well as consumption more in line with “traditional” foods such as the Mexican food group (e.g., tacos and quesadillas), potatoes and fried plantains, soup, legumes, and corn tortillas. The third pattern, the Eggs, Milk and Refined Grain pattern, was characterized by high intake frequency of refined grains, milk, sweetened milk, mayonnaise or margarine, and eggs. Each adolescent received a score for each of the three dietary patterns, with higher scores representing higher similarity to the pattern. Dietary pattern scores were then split into quartiles for analysis. See Appendix A for the average servings of food groups per day according to quartiles of dietary patterns (quartile 1 versus quartile 4).

### 2.2. Sleep Measures

At the end of each of the visits, adolescents were given an actigraph (ActiGraph GT3X+; ActiGraph LLC, Pensacola, FL, USA) to wear on their non-dominant wrist continuously for 7 days (97% of adolescents completed all 7 days in T1, and 83% completed all 7 days in T2). Adolescents also kept nightly sleep diaries to record bed and wake times. Nightly sleep parameters were estimated from the actigraphy and sleep diary data with the use of a fused least absolute shrinkage and selection operator (LASSO)-based calculator package developed in R Foundation for Statistical Computing, Vienna, Austria (R). From these data, weekday and weekend sleep duration (minutes), and weekday and weekend sleep midpoint (the median of sleep onset and wake time; reported in decimal hours) were obtained. As the primary endpoints of the study, sleep duration (weekday and weekend) and sleep midpoint (weekday and weekend) at follow-up, as well as change in sleep duration (weekday and weekend) and midpoint (weekday and weekend) from baseline to follow-up were used. It was decided a priori to conduct analyses separately for weekdays versus weekends because sleep habits may vary considerably, with typically more freedom to choose one’s sleep schedule on the weekend. It was hypothesized that these differences in sleep habits may result in different magnitudes of association. In a post-hoc analysis, the difference between weekend sleep midpoint and weekday sleep midpoint at follow-up, a marker of social jetlag, was also examined.

### 2.3. Covariates (From Baseline Visit)

Possible baseline confounders were selected based on a priori knowledge and included sex, age, school/work status, body mass index (BMI)-for-age Z scores, physical activity, screen time, smoking status (ever/never), maternal education, and household socioeconomic status. Participants were asked whether they were currently in school, and whether they were currently working. This information was used to create a three-category variable: currently in school, currently working (and not in school), and neither. Trained research assistants measured height in cm (Tonelli E120 A) and weight in kg (InBody230). BMI Z scores (BMIz) were calculated based on the World Health Organization reference [25]. Physical activity information was obtained from the actigraphs (described in detail here [26]) using Chandler’s vector magnitude cutoffs [27] and classified as moderate/vigorous hours/day. Total screen time was assessed with a questionnaire adapted for and validated in Mexican adolescents [28], and was divided into quartiles. Smoking behavior was self-reported with a single question “Have you ever tried smoking?” and was categorized dichotomously. Maternal education was reported by mothers at the original cohort enrollment visit and was classified into four categories: <8 years, 9–11 years, 12 years, and >12 years. Adolescents (with input from mothers when available) answered questions regarding current socioeconomic status. This questionnaire was created by the Asociación Mexicana de Agencias de Investigación de Mercados y Opinión Pública (Mexican Association of Marketing Research and Public Opinion Agencies) to evaluate household resources, including education of the household head, number of rooms in the house, number of vehicles owned, and ownership of particular appliances (e.g., microwave and washing machine). The index consists of seven categories ranging from A/B (highest SES) to E (lowest SES).

### 2.4. Statistical Analysis

First, to assess potential confounding factors, associations between covariates and sleep outcomes (duration and midpoint at T2, and change in duration and midpoint from T1 to T2, weekday and weekend) were examined by comparing the mean ± standard deviation (SD) of sleep outcomes across categories of covariates. Next, bivariate associations between dietary pattern quartiles and sleep measures were evaluated by estimating the means ± SD of sleep outcomes according to quartiles of each dietary pattern score. In multivariable analysis, separate linear regression models were run with continuous sleep measures as the outcome (each in a separate model) and indicator variables for quartiles of dietary pattern scores as the exposure (with the 1st quartile as the reference), adjusting for age, sex, screen time, and smoking. BMIz score was considered as a potential mediator and not included as a confounder. All other covariates were not strongly associated with the sleep outcomes and thus were not retained in final models. *p*-values for trend (*p*, trend) were obtained from linear regression models with sleep measures as the outcome and a variable representing ordinal categories of each dietary pattern. Interactions between dietary patterns and age were run to evaluate potential effect modification by age, but interactions were not apparent and are thus not reported. Finally, a post-hoc analysis that used linear regression to examine each of the diet patterns in relation to social jetlag (difference in weekday and weekend midpoints) at time 2 was performed. All analyses were conducted in Stata 14.0.

## 3. Results

Among the 458 participants included in the analysis, adolescents (46% male) were 14.4 ± 2.1 years of age at baseline, and the average follow-up length was 1.96 ± 0.4 years. At baseline, the overall average sleep duration was 516 ± 58 min (8.6 ± 1 h), while at follow-up it was 511 ± 68 min (8.5 ± 1.1 h). Overall average midpoint was 4:05 ± 1.3 and 4:28 ± 1.3 h, respectively. Considering weekdays and weekends separately, at baseline, the average weekday sleep duration was 505 ± 76 min (8.4 ± 1.3 h), and weekend sleep duration was 543 ± 80 (9.1 ± 1.3). Average weekday sleep midpoint was 3:50 AM ± 1.4 h, with a midpoint of 4:46 AM ± 1.3 h for weekends. At the second timepoint, average weekday sleep duration was 506 ± 84 min (8.4 ± 1.4 h), and weekend sleep duration was 527 ± 89 (8.8 ± 1.5). Average weekday sleep midpoint was 4:15 AM ± 1.4 h, with a midpoint of 5:01 AM ± 1.3 h for weekends.

Higher daily hours of moderate to vigorous activity (MVPA) were associated with shorter weekday sleep duration (Table 1), but no other associations were observed with sociodemographic or lifestyle characteristics. Male sex, older age, not being a student, more screen time, and ever having smoked cigarettes were associated with a later weekday sleep midpoint. In addition, adolescents with a normal BMI Z score had a later weekday sleep midpoint than adolescents who met the threshold for under or overweight/obese classifications.

From baseline to follow-up, there was, on average, no change in weekday sleep duration (−0.2 ± 101 min), while there was a 14.7 ± 105-min decline on weekends. Midpoint was delayed by 0.42 ± 1.54 and 0.28 ± 1.42 h on weekdays and weekends, respectively. Higher baseline moderate/vigorous physical activity was associated with reductions in weekday sleep duration. None of the baseline sociodemographic or lifestyle characteristics predicted changes in weekday sleep midpoint.

### 3.1. Associations with Weekday Sleep Characteristics

Bivariate and multivariable analysis revealed no associations between dietary patterns and weekday sleep duration (Table 2 and Table 3). However, higher consumption of the Plant-Based & Lean Proteins pattern (i.e., higher scores in this pattern) was associated with earlier weekday sleep midpoint at follow-up and less of a delay in sleep midpoint over time (Table 4). These associations persisted after accounting for sex, age, screen time, and smoking status, such that those in the highest quartile had a 0.45 h earlier midpoint (−0.81, −0.08; *p*, trend = 0.006) and a 0.39 h lower change in sleep midpoint over the follow-up period (−0.80, 0.02; *p*, trend = 0.02). The Eggs, Milk and Refined Grain pattern was also associated with an earlier sleep midpoint at follow-up. After accounting for confounders, adolescents in the highest quartile of this pattern had a 0.40 h (approximately 24 min) earlier sleep midpoint than those in the lowest quartile (−0.77, −0.04; *p*, trend = 0.03).

### 3.2. Associations with Weekend Sleep Characteristics

Higher consumption of the Plant-Based and Lean Proteins pattern was associated with shorter weekend sleep duration at follow-up, such that adolescents in the 4th quartile had a 0.5 h shorter sleep duration than those in quartile 1 (−0.9, −0.1; *p*, trend = 0.006; Table 3). Higher consumption of the Plant-Based and Lean Proteins pattern and the Eggs, Milk and Refined Grain pattern were associated with earlier weekend sleep midpoint at follow-up (*p*, trend 0.02 and 0.01, respectively; Table 4). In addition, higher consumption of the Plant-Based and Lean Proteins pattern was associated with a lower change in weekend sleep midpoint from baseline to follow-up (−0.36 with 95% CI −0.74, 0.03 in Q4 versus Q1; *p*, trend = 0.04).

### 3.3. Weekday–Weekend Differences

The Meat and Starchy pattern was not associated with any of the weekday or weekend sleep characteristics. However, post-hoc analysis showed that higher consumption of this pattern was associated with higher social jetlag (difference between weekend and weekday sleep midpoints) at follow-up (see Appendix A for bivariate association). Adolescents in the highest quartile of the Meat and Starchy pattern had a 0.32 higher adjusted difference in sleep midpoint on weekends compared to weekdays than adolescents in the lowest quartile (−0.01, 0.66; *p*, trend = 0.02; Appendix A).

## 4. Discussion

In this cohort of adolescents, the consumption of a healthy Plant-Based and Lean Proteins dietary pattern was associated with earlier sleep timing assessed 2 years later, as well as less of a phase delay (shift towards later timing) over 2 years. A pattern characterized by Eggs, Milk and Refined Grains (consistent with breakfast foods) was also associated with earlier sleep timing 2 years later, while a less healthy Meat and Starchy pattern was related to a higher social jetlag (difference in weekday and weekend timing) at follow-up. In general, diet patterns were not associated with sleep duration, although consumption of the Plant-Based and Lean Proteins pattern was associated with shorter weekend sleep duration at follow-up.

Overall, the study results showed that diet patterns that consisted of fruits, vegetables, lean proteins (including eggs from the Eggs, Milk, and Refined Grains pattern), and dairy were related to earlier sleep timing but not duration. The sleep timing results are in line with several cross-sectional adolescent studies, which showed that late bedtimes and/or wake times were associated with lower-diet-quality patterns containing more processed foods and added sugar [12,29,30]. However, the lack of an association with sleep duration is in contrast to several other studies which have found that lower-quality diet patterns or the intake of particular unhealthier foods are associated with short sleep duration in pediatric or adolescent populations [12,13,14]. One potential reason for this discrepancy could have to do with the fact that, in most pediatric and adolescent populations, sleep timing and duration are often highly inversely correlated; i.e., adolescents who go to bed late also have short sleep duration due to early school start times. However, in this Mexican population, sleep duration and timing are much less correlated since there is both a morning and an afternoon school shift, and thus not all adolescents have a constrained wake time. This fact is reflected in the typical weekday wake time of our sample, which is close to 8:30 AM, as well as the wide variability of weekday wake times (3:35 AM to 1:50 PM). Thus, our sample is uniquely situated to disentangle associations of sleep duration from sleep timing, and suggest that diet may be more closely connected with sleep timing than sleep duration. Another important aspect of the present study was the use of longitudinal data, such that the temporality between diet pattern adherence and prospective sleep could be established. Given that there are likely bidirectional relationships between sleep and diet, using prospective data represents the next step towards uncovering the extent to which a healthy dietary pattern could impact sleep. The ability to examine change in sleep timing over the adolescent period is also relevant biologically, because delays in sleep timing co-occur with pubertal changes. Further, the observed sleep midpoint effect sizes (between 20–30 min difference for the highest consumers versus the lowest) are within the realm of clinical significance, which is considered at least a 15-min difference in sleep timing (for adults) [31].

The Plant-Based and Lean Proteins pattern was associated with earlier sleep timing at follow-up, as well as a more favorable change in sleep timing from baseline to follow-up. These findings align with results from a recent intervention study among adults, in which participants whose diets became more anti-inflammatory (higher in fruits, vegetables, lean proteins and lower in added sugar and saturated fat) over a 3-month period experienced improvements in sleep onset latency [32]. Specific nutrients present in the Plant-Based and Lean Proteins pattern could also help explain the findings. Tryptophan, which is found in chicken and fish, is a precursor to melatonin [33], the hormone responsible for regulating circadian rhythms. Fish also contains docosahexaenoic acid, an omega 3 fatty acid that has been associated with earlier sleep timing within this cohort [16]. The mechanism by which docosahexaenoic acid impacts sleep may also be related to melatonin. Foods that promote melatonin production may be especially important during the adolescent period, during which there are changes in the nightly release of melatonin (which occurs much later at night than at other times of the lifespan). The Plant-Based and Lean Proteins pattern was also associated with shorter sleep duration on the weekends, which may seem to be in the unexpected direction. However, shorter sleep duration on weekends may be an indirect marker of better sleep health overall, as long sleep duration on weekends (aka “catch-up” sleep) can indicate overcompensating for poor quality sleep or short sleep duration on weekdays.

The Eggs, Milk and Refined Grain pattern was also associated with earlier sleep timing at follow-up, although it was not related to more favorable change in sleep over time. The Eggs, Milk and Refined Grain pattern is composed of foods commonly consumed for breakfast, so our results may reflect the impact of habitual breakfast on regulating circadian rhythms and promoting earlier bedtimes [34]. Nonetheless, this hypothesis cannot be fully evaluated as neither the timing nor composition of breakfast was asked about within the questionnaire. As an alternative explanation, both milk and eggs (a source of tryptophan), foods present within the pattern, may help promote melatonin production [35].

The Meat and Starchy dietary pattern, which was characterized by lower-quality and more heavily processed foods, was not associated with any of the primary outcomes, but it was associated with a higher social jetlag at follow-up. Other cross-sectional literature found links between higher social jetlag and worse-quality diets [36], although it is typically thought about in the reverse direction (i.e., social jetlag affecting diet). Further, animal studies and small human experimental studies have shown that high-saturated-fat diets may impact circadian rhythms [37,38,39] and other studies have found that red meat is associated with lower sleep duration and worse sleep quality [40], sleep characteristics that are often concomitant with social jetlag.

This study had several key strengths. First was the prospective nature of the study, with repeated timepoints for sleep outcomes. Second, the sleep data were recorded objectively with the use of an actigraph. Third, the dietary pattern approach to characterize diet allows a more comprehensive examination of diet, which is also relevant for public health recommendations. In addition, the data-driven approach to identify dietary patterns allows the identification of naturally occurring patterns within the population. There were also limitations. First, diet was self-reported, which can be subject to measurement error, both random and systematic. Second, dietary habits were assessed over the previous seven days, which may not be reflective of long-term intake. In addition, the timing of food intake was not assessed, which could have an important impact on circadian rhythms. Third, as in any observational study, causality cannot be assumed. It is pertinent to consider that the findings could be reflective of overall lifestyles and family environments, such that adolescents who have healthier environments for diet likely also have healthier environments for sleep. Household socioeconomic status and other individual lifestyle factors (screen time, smoking) were included as confounders, and their adjustment did not alter the findings, although there may still be residual confounding due to other household-level factors. Fourth, generalizability may be limited to adolescents within urban Latin American settings. Similarly, because the dietary patterns were data-driven, they may not be comparable across different populations.

In summary, within this cohort of Mexican adolescents, there was evidence that the consumption of a healthy plant-based diet, as well as a breakfast-type dietary pattern, were associated with earlier sleep timing measured two years later. The plant-based diet was also associated with smaller phase delays in sleep timing over time. In contrast, a meat and carbohydrate-based diet pattern was associated with higher social jetlag. These findings support the potential benefits of a healthy diet alongside other sleep hygiene guidelines for helping to achieve and maintain healthy sleep among adolescents.

## Figures and Tables

**Table 1 nutrients-12-02305-t001:** Sleep characteristics at T2 and change in weekday sleep characteristics from T1 to T2, stratified by baseline sociodemographic and lifestyle characteristics.

Baseline Sociodemographic and Lifestyle Characteristics	*n*	Sleep Duration at T2, Minutes ± SD	Change in Sleep Duration T1 to T2 (Min)	Sleep Midpoint at T2, Decimal Hours ± SD	Change in Sleep Midpoint T1 to T2 (H)
Sex					
Male	211	504 ± 82	0.55 ± 100.3	4.4 ± 1.5	0.5 ± 1.5
Female	247	504 ± 86	−0.86 ± 102.7	4.1 ± 1.3	0.4 ± 1.6
*p*-value ^2^		0.38	0.87	0.05	0.43
Age group, years					
9.5 to <12	80	494 ± 74	−18.8 ± 76.7	3.7 ± 1.3	0.4 ± 1.4
12 to <14	142	511 ± 79	0.6 ± 95.3	4.1 ± 1.5	0.5 ± 1.5
14 to <16	84	509 ± 96	2.7 ± 106.3	4.4 ± 1.3	0.6 ± 1.4
16 to 18	152	505 ± 87	7.2 ± 114.6	4.5 ± 1.4	0.2 ± 1.6
*p*-value ^2^		0.19	0.28	0.0003	0.26
School/work status					
Currently enrolled in school	427	506 ± 85	2.8 ± 101	4.2 ± 1.4	0.5 ± 1.5
In workforce (not in school)	8	488 ± 80	−8.3 ± 99	4.8 ± 1.4	0.2 ± 1.1
Neither	19	509 ± 79	−47 ± 110	5.0 ± 1.5	−0.4 ± 1.6
*p*-value ^2^		0.77	0.09	0.05	0.16
BMI *Z* score					
<0	46	508 ± 82	−20.1 ± 101.2	3.9 ± 1.2	0.3 ± 1.3
0 to <1	112	516 ± 89	1.3 ± 98.0	4.6 ± 1.5	0.6 ± 1.5
1 to <2	125	503 ± 85	4.6 ± 118.6	4.2 ± 1.5	0.4 ± 1.7
≥2	173	500 ± 80	0.8 ± 90.5	4.1 ± 1.4	0.4 ± 1.5
*p*-value ^2^		0.40	0.31	0.01	0.71
Maternal education, years (y)					
9 y or less	50	521 ± 75	15.5 ± 111.9	4.4 ± 1.5	0.6 ± 1.4
10 to <12 y	183	510 ± 84	0.8 ± 99.5	4.3 ± 1.4	0.4 ± 1.6
12 y	152	498 ± 86	−9.6 ± 100.6	4.1 ± 1.4	0.4 ± 1.5
>12 y	68	504 ± 88	8.6 ± 102.0	4.3 ± 1.6	0.3 ± 1.7
*p*-value ^2^		0.35	0.54	0.62	0.50
Socioeconomic status					
A/B, C+ or C	116	514 ± 88	13.9 ± 111.4	4.3 ± 1.6	0.4 ± 1.6
C−	108	491 ± 77	−14.6 ± 86.9	4.1 ± 1.2	0.3 ± 1.6
D+	114	510 ± 85	3.1 ± 94.0	4.4 ± 1.4	0.6 ± 1.5
D or E	120	506 ± 85	−4.0 ± 109.3	4.3 ± 1.5	0.4 ± 1.5
*p*-value ^2^		0.20	0.25	0.44	0.54
Moderate/Vigorous Activity (Min/day), quartiles					
Q1, 15.5–59.3	114	517 ± 92	6.3 ± 103.9	4.4 ± 1.5	0.4 ± 1.6
Q2, 59.4–75.3	113	516 ± 88	14.9 ± 111.8	4.3 ± 1.5	0.4 ± 1.6
Q3, 75.6–97.2	113	495.8 ± 81	−2.1 ± 100.5	4.2 ± 1.4	0.4 ± 1.4
Q4, 97.6–216.3	113	494 ± 72	−20.7 ± 86.9	4.1 ± 1.4	0.5 ± 1.6
*p*-value ^2^		0.01	0.02	0.27	0.94
Screen time, quartiles					
Q1, 1 to <23 h/week (h/week)	121	503 ± 78	−7.3 ± 104.2	4.0 ± 1.4	0.3 ± 1.5
Q2, 23 to <33 h/week	108	512 ± 90	13.9 ± 101.3	4.3 ± 1.5	0.7 ± 1.6
Q3, 33 to <48.5 h/week	113	500 ± 87	−2.0 ± 100.8	4.2 ± 1.4	0.3 ± 1.5
Q4, 48.5 to 116 h/week	116	508 ± 82	−4.1 ± 99.4	4.5 ± 1.3	0.3 ± 1.5
*p*-value ^2^		0.71	0.64	0.06	0.24
Ever smoked cigarettes					
No	345	506 ± 85	−1.5 ± 101.9	4.1 ± 1.4	0.4 ± 1.5
Yes	109	507 ± 81	6.9 ± 100.2	4.7 ± 1.5	0.4 ± 1.6
*p*-value ^2^		0.71	0.46	0.0005	0.95

^2^ From Kruskal–Wallis tests.

**Table 2 nutrients-12-02305-t002:** Bivariate associations between baseline dietary patterns and prospective weekday sleep characteristics.

Baseline Dietary Patterns	*n*	Sleep Duration at T2, Minutes ± SD	Change in Sleep Duration T1 to T2 (Min)	Sleep Midpoint at T2, Minutes ± SD	Change in Sleep Midpoint T1 to T2 (H)
Plant-based & Lean Proteins					
Q1	115	504 ± 91	−7.3 ± 109.3	4.5 ± 1.4	0.5 ± 1.6
Q2	114	512 ± 87	4.7 ± 99.0	4.3 ± 1.4	0.7 ± 1.5
Q3	115	506 ± 81	−0.3 ± 98.3	4.1 ± 1.4	0.3 ± 1.6
Q4	114	501 ± 78	2.2 ± 99.8	4.0 ± 1.4	0.2 ± 1.5
*p*, trend		0.81	0.94	0.03	0.04
Meat & Starchy					
Q1	115	503 ± 93	−9.0 ± 103.6	4.2 ± 1.2	0.3 ± 1.4
Q2	114	512 ± 79	−0.5 ± 93.2	4.4 ± 1.5	0.4 ± 1.6
Q3	115	505 ± 78	1.7 ± 103.8	4.2 ± 1.5	0.5 ± 1.6
Q4	114	502 ± 86	7.0 ± 105.5	4.2 ± 1.4	0.4 ± 1.5
*p*, trend		0.71	0.72	0.29	0.78
Eggs, Milk & Refined Grain					
Q1	115	510.6 ± 83	−3.3 ± 100.5	4.5 ± 1.5	0.4 ± 1.6
Q2	114	503 ± 84	10.3 ± 98.5	4.3 ± 1.4	0.5 ± 1.5
Q3	115	503 ± 81	4.1 ± 100.7	4.2 ± 1.5	0.6 ± 1.6
Q4	114	506 ± 89	−11.9 ± 106.1	4.0 ± 1.4	0.1 ± 1.5
*p*, trend		0.73	0.33	0.05	0.09

**Table 3 nutrients-12-02305-t003:** Multivariable associations between baseline dietary patterns and prospective sleep duration.

Baseline Dietary Patterns	*n*	Weekday Sleep Duration at T2, Adjusted Difference (Min) ^1^	Weekend Sleep Duration at T2, Adjusted Difference (Min) ^1^
Plant-based & Lean Proteins			
Q1	115	Reference	Reference
Q2	114	6.43 (−16.24, 29.09)	3.55 (−19.49, 26.59)
Q3	115	0.08 (−22.34, 22.50)	−12.03 (−34.93, 10.86)
Q4	114	−3.58 (−26.34, 19.18)	−29.80 (−53.19, −6.41)
*p*, trend		0.64	0.01
Meat & Starchy			
Q1	115	Reference	Reference
Q2	114	9.38 (−13.12, 31.88)	6.39 (−16.76, 29.54)
Q3	115	1.50 (−20.87, 23.87)	−1.43 (−24.37, 21.52)
Q4	114	−1.64 (−24.04, 20.76)	7.31 (−15.67, 30.28)
*p*, trend		0.72	0.70
Eggs, Milk & Refined Grain			
Q1	115	Reference	Reference
Q2	114	−6.11 (−28.71, 16.49)	5.11 (−18.06, 28.28)
Q3	115	−7.83 (−30.31, 14.64)	15.73 (−7.50, 38.97)
Q4	114	−2.58 (−25.17, 20.02)	7.22 (−15.97, 30.41)
*p*, trend		0.79	0.39

^1^ Adjusted for sex, age, baseline screen time, and baseline smoking status.

**Table 4 nutrients-12-02305-t004:** Multivariable associations between baseline dietary patterns and prospective sleep midpoint.

Baseline Dietary Patterns	*n*	Weekday Sleep Midpoint at T2, Adjusted Difference (H) ^1^	Change in Weekday Sleep Midpoint T1 to T2, Adjusted Difference (H) ^1^	Weekend Sleep Midpoint at T2, Adjusted Difference (H)	Change in Weekend Sleep Midpoint T1 to T2, Adjusted Difference (H) ^1^
Plant-based & Lean Proteins					
Q1	115	Reference	Reference	Reference	Reference
Q2	114	−0.12 (−0.48, 0.25)	0.15 (−0.26, 0.55)	−0.15 (−0.50, 0.21)	0.02 (−0.36,0.40)
Q3	115	−0.38 (−0.74, −0.03)	−0.27 (−0.67, 0.13)	−0.35 (−0.70, 0.00)	−0.18 (−0.55, 0.20)
Q4	114	−0.45 (−0.81, −0.08)	−0.39 (−0.80,	−0.39 (−0.75, −0.03)	−0.36 (−0.74, 0.03)
*p*, trend		0.006	0.02	0.02	0.04
Meat & Starchy					
Q1	115	Reference	Reference	Reference	Reference
Q2	114	0.23 (−0.13, 0.59)	0.04 (−0.37, 0.44)	0.02 (−0.33, 0.38)	−0.08 (−0.47, 0.30)
Q3	115	−0.07 (−0.43, 0.29)	0.13 (−0.27, 0.53)	0.03 (−0.32, 0.39)	0.14 (−0.23, 0.52)
Q4	114	−0.01 (−0.37, 0.35)	0.06 (−0.34, 0.46)	0.28 (−0.07, 0.63)	0.22 (−0.16, 0.60)
*p*, trend		0.58	0.66	0.14	0.15
Eggs, Milk & Refined Grain					
Q1	115	Reference	Reference	Reference	Reference
Q2	114	−0.12 (−0.48, 0.25)	0.06 (−0.34, 0.47)	−0.18 (−0.53, 0.18)	−0.12 (−0.50, 0.26)
Q3	115	−0.20 (−0.56, 0.16)	0.11 (−0.29, 0.51)	−0.17 (−0.53, 0.19)	−0.06 (−0.44, 0.32)
Q4	114	−0.40 (−0.77, −0.04)	−0.28 (−0.69, 0.12)	−0.50 (−0.85, −0.14)	−0.14 (−0.52, 0.25)
*p*, trend		0.03	0.22	0.01	0.58

^1^ Adjusted for sex, age, baseline screen time, and baseline smoking status.

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
