# Peer review of "Dietary Patterns in Relation to Prospective Sleep Duration and Timing among Mexico City Adolescents"

_nutrients, 2020, doi:10.3390/nu12082305_

Round 1

Reviewer 1 Report

Dear Authors,

The study investigated adolescent consumption of three dietary patterns and their association with sleep duration and sleep timing.  Results showed that adolescents consuming higher amounts of plant-based and lean proteins had earlier sleep timing compared to adolescents that consumed less plant-based and lean proteins. Adolescents that consumed greater amounts of eggs, milk and refined grains also slept earlier than those consuming lower amounts. Adolescents with a higher meat and starchy pattern had greater social jet-lag.

I would like to congratulate the authors on this paper.  The manuscript was very well written and it was a pleasure to read.  The introduction is very well written and covers a broad range of literature to support the rationale for the study. The design of the study is very good, including a novel two year follow up. The statistical analysis is also appropriate for the design. The discussion is balanced, highlighting plausible explanations for the results and limitations of the research.

Accordingly, I have only two minor comments for consideration.

  • In my view, the term ’adherence’ suggests commitment to a programme. This gives the impression (particularly in the abstract) that the adolescents were placed on a dietary programme. It might be useful to perhaps replace this term.
  • In the results section, line 96 – it would be useful to include a summary of the sleep parameters from the first time point.

Reviewer 2 Report

Re: Dietary patterns in relation to prospective sleep duration and timing among adolescents.

The relationship between sleep and diet is of emerging interest to nutritional research and this study presents data for adolescents – a key life stage in development of lifestyle behaviours.  There are a number of areas (detailed below) where extra clarity and further explanation is required.

Title: 

Add cohort / location of study.

Abstract:

Line 19 - state a posteriori diet patterns

General - report fraction of hours or minutes.

Introduction:

General - easier to read if 'adolescents' rather than 'youth' used.

Line 34/35 - add definition of adolescents - this does vary, but good to give typical age range.

Line 48-50 -  brief mention of different diet score approaches, a priori/ a posteriori.

Line 59/60 - reference needed around evidence of foods and melatonin secretion promotion.

Methods:

Line 81/84 - is this FFQ validated / tested in this population group?

Line 87/88 - the a posteriori scores previously reported/ published are the primary exposure of the study.  More detail is needed - this could be added to the supplementary material.  Line 63 refers to 'healthier' diet pattern - the 'plant and lean protein' pattern? it would be good to see how the nutrient intake / food group intakes correspond to adherence to these patterns and how they compare.  This would help interpretation of the results.

Line 118/119 - state method of classification of PAL.

The age range of 9.5 yrs to 18 years is highly heterogenous.  This was taking into account in adjustments.  However, were any stratified analyses run as the 16-18year age group may be very different from early/pre teens?  You may want to consider comparing the older age group to younger?

Results:

Tables: sleep duration outcomes - are these all in minutes? the units in the tables are not clear.

Figure 1 - needs to include sample size, r and p value. Is this analysis adjusted? As this is post hoc the figure could go in supplementary.  Not clear on rationale why this analysis only ran for meat and starchy group.

Discussion:

Line 251 & 267: reference needed to support this observation in humans (dietary tryptophan and melatonin).

A limitation is that time of intakes were not reported - potentially important impact on circadian rhythms and potential confounding factor in the reported results.  This needs to be discussed.

To add strength/limitation around application of data driven dietary scores.

Are the differences in times observed likely significant in terms of health outcomes?
